# Japanese Encephalitis Vaccine in Low- and Middle-Income Countries (LMICs): A Narrative Review of Efficacy, Effectiveness, Safety, Cost, and Policy

**DOI:** 10.3390/vaccines13101038

**Published:** 2025-10-08

**Authors:** Eufrasia Ine Pilihanto, Btari Kalisha Nyratri, Muhammad Dafrizal Firdaus, Rano Kurnia Sinuraya

**Affiliations:** 1Faculty of Pharmacy, Universitas Padjadjaran, Sumedang 45363, West Java, Indonesia; eufrasia22001@mail.unpad.ac.id (E.I.P.); btari22001@mail.unpad.ac.id (B.K.N.); muhammad22147@mail.unpad.ac.id (M.D.F.); 2Department of Pharmacology and Clinical Pharmacy, Faculty of Pharmacy, Universitas Padjadjaran, Sumedang 45363, West Java, Indonesia; 3Center of Excellence for Pharmaceutical Care Innovation, Universitas Padjadjaran, Bandung 40132, West Java, Indonesia

**Keywords:** *Japanese encephalitis*, vaccination, low- and middle-income countries, cost-effectiveness, policy implementation

## Abstract

Japanese Encephalitis (JE) is a mosquito-borne viral infection that causes acute brain inflammation. First identified in Japan in 1871, the disease gained renewed global attention in 2025 after emerging in a non-endemic region, raising significant healthcare concerns. Vaccination remains the most effective strategy for preventing outbreaks. However, low- and middle-income countries (LMICs) face considerable challenges in implementing vaccination programs due to geographical, economic, and regulatory barriers. Most existing studies on JE vaccines (JEVs) have been conducted in higher-income countries, leaving critical gaps in data on efficacy and safety in LMIC settings. Furthermore, uncertainties surrounding cost-effectiveness make funding decisions more complex. This narrative review evaluates the current evidence on JE vaccination in LMICs, based on a literature search in PubMed and ScienceDirect covering 2005–2025. The review examines vaccine efficacy, safety, cost-effectiveness, and policy implementation. Findings show that JEVs demonstrate high efficacy and strong safety profiles, with mild adverse effects, most commonly fever. The live attenuated SA 14-14-2 vaccine (LAJEV) is particularly cost-effective, offering substantial economic benefits by reducing healthcare expenditures in endemic regions. To ensure sustainability, vaccination programs in LMICs require tailored policies and targeted financial support. Policy frameworks must be adapted to local contexts, enabling focused, effective, and equitable implementation.

## 1. Introduction

Japanese Encephalitis (JE) is an infectious neurological disease characterized by acute inflammation of the central nervous system. It is the most severe form of viral encephalitis caused by the Japanese Encephalitis Virus (JEV), a member of the *Flaviviridae* family. The transmission cycle of JEV involves amplifying hosts, such as aquatic wading birds and swine, while mosquitoes—particularly *Culex* species—serve as vectors, transferring the virus between hosts [1,2]. JE is endemic to tropical regions of Southeast Asia but has expanded to other parts of Asia, as well as to Australia, Africa, and Europe. In endemic regions, most cases occur in children under 15 years of age, as adults typically acquire immunity through prior exposure. However, in newly affected areas, both children and adults lack pre-existing antibodies, making them highly vulnerable [3,4,5,6].

JEV is phylogenetically classified into five genotypes (GI–GV), based on nucleotide variation in the E-protein gene [7]. Current vaccines, derived from GIII strains, induce neutralizing antibodies against genotypes I–IV, providing cross-protection [8]. Genotype V, the oldest lineage, was first isolated in Malaysia in 1952. After nearly six decades without detection, GV re-emerged in mosquitoes in Tibet, China, in 2009 and later in South Korea in 2011. Evidence suggests that GIII-based vaccines may induce reduced neutralization against genotype V compared with other lineages; however, the clinical significance of this finding remains uncertain and warrants further investigation [8,9].

The early stages of JE often present with mild, nonspecific symptoms such as fever, chills, diarrhea, headache, vomiting, and fatigue. As the disease progresses, severe neurological manifestations such as altered mental status, focal deficits, and movement disorders may develop [10]. Because JE closely resembles other acute neurological infections, laboratory confirmation is essential for accurate diagnosis [11]. Most infections are subclinical, with an estimated symptomatic-to-asymptomatic ratio of 1:25–1000 [1]; although the WHO estimates this at approximately 1:250. Among symptomatic cases, the case-fatality rate can reach 30% in children under 15. Half of survivors in this age group suffer from irreversible neurological sequelae such as epilepsy, stroke, altered mental status, coma, or stupor [12]. JE has been reported to cause particularly high disability-adjusted life year (DALY) losses, with some analyses ranking it above other mosquito-borne viral diseases [13].

The first documented clinical case of JE occurred in Japan in 1871. Genetic analyses indicate that the oldest lineage (Indonesian isolate JKT6468) originated in the Indonesia–Malaysia region, with other genotypes found widely across Southeast Asia, including Thailand, Cambodia, and Korea [14]. EV has expanded beyond its traditional range, with outbreaks reported in Europe (2010), Africa (2016), and mainland Australia (2021–2022) [15]. Genotypes I–V have been reported in diverse regions including Indonesia, Malaysia, Australia, Cambodia, Papua New Guinea, Taiwan (China), the Philippines, Vietnam, Laos, Myanmar, Japan, South Korea, India, Sri Lanka, Nepal, Italy, and Angola [16]. Outbreak risk in endemic regions is heightened by tropical climates with seasonal peaks during the rainy season, coupled with rural pig farming that serves as an efficient viral reservoir [17].

The annual incidence of clinical JE varies across endemic countries and within regions, exceeding 10 per 100,000 during outbreaks. A systematic review and modeling study estimated approximately 100,000 clinical cases globally each year (95% CI: 61,720–157,522), leading to about 25,000 deaths (95% CI: 14,550–46,031) [18,19]. In 2024, Vietnam reported the highest number of cases among low- and middle-income countries (LMICs), with 134 cases (Figure 1). However, WHO data from 2001 to 2024 show that India recorded the largest cumulative burden, with 22,892 cases [20].

In March 2022, Australia declared the JE outbreak a Communicable Disease Incident of National Significance following human cases across multiple states [21]. While vaccination programs have reduced incidence in children, cases are increasing among adults and non-endemic travelers. Climate change is thought to influence mosquito populations and alter transmission dynamics [10,22]. Evidence also suggests that JEV may have reached northern Australia via wind-dispersed mosquitoes from New Guinea [17].

### Vaccination Gaps in LMICs

WHO strongly recommends vaccination against JE in endemic regions across all age groups, though schedules and target populations vary by vaccine type (Table 1). For children, vaccination typically begins at 9 months, with two doses given 1–2 years apart. Adults usually require a single dose [6]. Four main JEV types are currently available: mouse brain–derived inactivated vaccines (IMBV), cell culture–derived inactivated vaccines (JE-VC), live attenuated vaccines (LAJEV), and genetically engineered live-attenuated chimeric vaccines (JE-CV) [3,5].

In addition to existing vaccines, several novel candidates are under development, including recombinant protein-based vaccines, poxvirus-based vaccines, and plasmid DNA-based vaccines [6]. The JE-VC, a widely used cell culture–derived inactivated vaccine, is produced in Austria and France [27]. WHO recommends that countries where JE is a significant public health concern incorporate JE vaccination into their national immunization schedules. Some LMICs, including India, Sri Lanka, and Cambodia, have already introduced JE vaccination programs with support from the Global Alliance for Vaccines and Immunization (GAVI), which works to expand vaccine access in resource-limited settings [18,28,29].

Despite the availability of effective vaccines, JE control in LMICs remains difficult due to multiple barriers. Limited healthcare infrastructure often delays diagnosis and treatment, particularly in rural areas where transportation to medical facilities is restricted. Economic constraints also prevent many low-income families from accessing vaccination and adequate medical care. Moreover, the high cost of vaccines, coupled with the need for multiple doses, places additional strain on already fragile health systems [30]. Vaccine delivery in LMICs faces further obstacles, including inadequate infrastructure for conducting clinical trials, insufficient regulatory capacity, and a lack of sustainable financing mechanisms [31]. These systemic issues hinder widespread adoption and long-term program sustainability.

Data on the efficacy, effectiveness, and safety of JE vaccines in LMICs are limited, as most research has been conducted in high-income countries. In endemic regions, adult efficacy studies are particularly scarce, since many adults have naturally acquired immunity during childhood. The cost-effectiveness of vaccination in resource-limited settings also remains uncertain, complicating government decisions on prioritization and investment.

In addition, variations in policy support and implementation across LMICs contribute to inconsistent vaccination coverage. A comprehensive literature review is needed to assess the performance, safety, cost-effectiveness, and policy adoption of JE vaccines specifically in LMICs. Such an assessment would help identify barriers to vaccine access and guide strategies to strengthen immunization programs.

## 2. Materials and Methods

A literature search was conducted in PubMed and ScienceDirect for studies published between 2005 and 2025. The search terms included “Japanese Encephalitis Vaccines,” “efficacy,” “effectiveness,” “safety,” “cost,” and “policy.” Studies were eligible if they met the following criteria: (1) clinical trials, observational studies, or health policy documents focusing on JEV in LMICs; (2) reported outcomes related to vaccine effectiveness, efficacy, safety, cost, or policy; and (3) were available in full text and published in English. Titles and abstracts of all identified records were independently screened for relevance by three reviewers (EIP, BKN, and MDF). Full texts of potentially eligible studies were then retrieved and assessed independently by the same reviewers using the inclusion and exclusion criteria. Discrepancies were resolved through discussion. From each study, the following data were extracted: author(s), year, country, study design, vaccine type, study population, outcomes assessed, and key findings. Because the included studies differed in design, vaccine type, outcome measures, and follow-up duration, a narrative synthesis was conducted to summarize the findings.

## 3. Results

From 2005 to 2025, our search of PubMed and ScienceDirect identified 88 publications on JE vaccine efficacy and effectiveness, of which 15 studies conducted in LMICs met the inclusion criteria. For safety, 65 publications were retrieved, and 16 were eligible after screening. The cost-effectiveness search yielded 86 publications, with 8 meeting criteria in LMIC settings. For policy, 13 articles were retrieved, but none originated from LMICs. However, two additional policy-related studies were identified during the detailed review of efficacy, effectiveness, safety, and cost-effectiveness articles.

### 3.1. Efficacy and Effectiveness

Vaccine efficacy is typically assessed through controlled clinical trials by comparing the proportion of vaccinated individuals who develop the outcome of interest (usually disease) with those who receive a placebo. Efficacy reflects how much a vaccine reduces disease risk under trial conditions. In contrast, vaccine effectiveness describes how well a vaccine performs in real-world settings. Although clinical trials include participants across diverse age groups, sexes, ethnicities, and health conditions, they cannot perfectly represent the entire population [32,33].

Clinical evaluation of JEV is challenging due to the low incidence of the disease and the ethical constraints of placebo-controlled trials in regions where licensed vaccines are available. These challenges necessitate the use of surrogate immunological markers acceptable to regulators for licensure. The plaque reduction neutralization test (PRNT_50_) has become the gold standard for assessing protection, as it quantifies neutralizing antibody titers (NATs) required to achieve 50% viral neutralization. A PRNT_50_ titer ≥ 1:10 is widely recognized as the threshold for seroprotection, given its correlation with protection, including cross-genotype efficacy [34]. Seroprotection is defined as the proportion of individuals achieving antibody levels considered sufficient to prevent disease. However, low NATs do not always indicate loss of protection, as immune memory can persist even after antibody decline, allowing rapid recall responses upon re-exposure. This underscores the value of including memory response data in regulatory submissions. T cell–mediated immunity, especially CD8^+^ cytotoxic T lymphocytes, also plays a role in viral clearance and limiting neuroinvasion, highlighting the importance of cellular responses in long-term protection.

Seroconversion, defined as the development of detectable antibodies after vaccination [35], is commonly evaluated in non-inferiority trials, along with geometric mean titers (GMTs). Regulatory consensus supports a seroconversion rate (SCR) benchmark of ≥75% in the comparator vaccine group, with a non-inferiority margin of −10% for the lower bound of the 95% confidence interval [36]. Table 2 summarizes the efficacy and effectiveness of JEVs used in LMICs.

#### 3.1.1. LAJEV Efficacy and Effectiveness Among Children and Infants (6 Months–18 Years)

A cross-sectional study by Tun and colleagues in Myanmar reported that 100% of children tested positive for anti-JE Immunoglobulin G (IgG) after a single dose of LAJEV. Six months later, 87% maintained neutralizing antibodies (NATs). The lowest IgG geometric mean titers (GMTs) were observed in the 5–7-year age group, while the highest were in the 14–15-year age group, suggesting an age-related increase in GMTs [37].

In Bangladesh, Zaman and colleagues conducted a phase 4 clinical study evaluating antibody responses following a booster dose of LAJEV. Among 560 children vaccinated at 10–12 months of age, 524 received a booster 3–4 years later. Seroprotection was achieved in 91.4% of children by day 7 post-booster, rising to 98.1% by day 28. The GMT increased from 6 at baseline to 105 on day 7 and 167 on day 28 [38].

A study in Sri Lanka by Wijesinghe and colleagues assessed children who had previously received at least two doses of IMBV. After LAJEV vaccination, these children showed significant increases in antibody levels, with GMTs remaining high for up to one year post-vaccination. At 28 days post-vaccination, 52.8% of 2-year-olds and 40.4% of 5-year-olds achieved a ≥4-fold increase in NATs, including some who were already seropositive at baseline [39].

In an earlier open-label clinical trial in Sri Lanka, Wijesinghe demonstrated that LAJEV co-administered with the measles vaccine (MV) in 9-month-old infants achieved a 90.7% seropositivity rate within 28 days post-vaccination. At one year, seropositivity remained at 87.4%, although GMTs declined from 111 to 76 over time [40].

In the Philippines, Gatchalian and colleagues reported that LAJEV given with MV provided comparable levels of seroprotection against JE, regardless of whether the vaccines were administered together or separately [41]. Conversely, Capeding and colleagues observed lower seroprotection rates, with 72.3% for co-administration and 68.2% for sequential administration at 28 days post-vaccination. The authors suggested that these differences might reflect variability in neutralizing antibody assays [42].

#### 3.1.2. JE-CV Efficacy Among Children

An open-label trial in Thailand by Janewongwirot and colleagues evaluated JE-CV as a booster following an initial LAJEV dose in healthy children aged 1–5 years. By day 28, PRNT_50_ titers against JE-CV increased significantly from 162 to 6934, with approximately 96% of participants achieving seroprotection. However, PRNT_50_ titers against the wild-type JE virus (Beijing strain) and LAJEV were lower than those for JE-CV [41]. Similarly, a clinical trial by Feroldi and colleagues conducted in the Philippines and Thailand assessed both locally manufactured and U.S.-produced JE-CV in children aged 12–18 months. In Thailand, 95.5% of participants achieved at least a 14-fold increase in antibody titers within 28 days, and GMTs rose more than four-fold, from 16.6 to 241, with comparable results across both countries and vaccine lots [43].

#### 3.1.3. Inactivated JEV Efficacy Among Children

A clinical study in Thailand by Chanthavanich and colleagues evaluated JE-VC in 152 healthy children aged 1–2 years, without a control arm, and assessed immunogenicity through SCRs and GMTs at four weeks after the second and third (booster) doses. The results showed high efficacy, with 100% SCRs and GMTs rising from 150 after the second dose to 621.7 after the booster [44]. Chokephaibulkit and colleagues examined IMBV in 21 HIV-infected and 101 HIV-uninfected children aged ≥12 months born to HIV-infected mothers, finding that GMTs were lower in HIV-infected children, although seroconversion still occurred in 83% at one month and 94% at three months, compared with 99% and 100% in uninfected children, respectively; all participants had NATs < 10 against the Beijing strain before vaccination [45]. In India, a phase 4 multicenter, open-label randomized controlled trial by Vadrevu and colleagues reported that JENVAC achieved an 81.7% SPR at day 360 post-vaccination, whereas LAJEV showed lower seroprotection (47.9%) [46]. In Vietnam, Marks and colleagues reported 92.9% efficacy for IMBV in children aged 1–5 years in a case–control study comparing confirmed JE cases with healthy controls [47]. Similarly, a randomized, open-label phase II study in Bangalore, India, by Koltenbobock and colleagues found that 95.7% of children receiving a 3-µg dose and 95.2% of those receiving a 6-µg dose of IXIARO^®^ achieved seroconversion by day 56 post-vaccination, with both doses proving effective compared with the standard JEV used locally (JenceVac) [48].

#### 3.1.4. JEV Efficacy and Effectiveness Among Adults

In LMICs, evidence on JEV effectiveness among adults remains limited. In India, Khan and colleagues conducted an observational study involving 1075 healthy adults aged ≥15 years who received LAJEV. Prior to vaccination, 9.3% of participants were seronegative, 24.5% were moderately seropositive, and 66.2% were strongly seropositive, with GMTs ranging from 6.22 to 373.82. Among seronegative individuals, 85.5% achieved seroconversion by day 28, with GMTs showing a four-fold increase and remaining above the protective threshold for 12 months; overall, 95% of participants remained seroprotected at one year [49]. In Vietnam, a clinical trial of JE-VC in adults aged 12–60 years demonstrated a 98.5% seroprotection rate by day 28 post-vaccination, with titers ≥ 10 sustained during follow-up [50]. Collectively, these findings indicate that both LAJEV and JE-VC are highly effective in adult populations, achieving seroprotection rates above 85–95% in LMIC settings.

**Table 2 vaccines-13-01038-t002:** Efficacy and Effectiveness of JEV.

No	Country Site	Participant	Type of JE Vaccine	Type of Study	Result	Ref.
**Efficacy**						
1	Bangladesh	560 children aged 10–12 months	LAJEV	Phase 4, open-label clinical study	GMT increased from 6 at baseline to 105 (day 7) and 167 (day 28) post-booster	[38]
2	Sri Lanka	Children aged 2–5 years	LAJEV	Open-label, single arm trial	Seroconversion achieved by 53.7% of 2-year-olds and 40.8% of 5-year-olds after booster vaccination	[39]
3	Sri Lanka	Infants aged 9 months (±2 weeks)	LAJEV	Open-label, non-randomized, single-arm trial	The GMT of NATs was 111 within 28 days post-vaccination	[40]
4	Thailand	50 healthy children aged 1–5 years (64% male)	JE-CV	Open-label clinical trial	PRNT_50_ titers increased from 162 to 6934 against JE-CV and from 58 to 979 against wild-type virus following a booster dose	[41]
5	Philippines	628 healthy children aged 9–12 months	LAJEV	Randomized clinical trial	Seroprotection rates were 72.3% (co-administered with MV) and 68.2% (sequential administration) at 28 days post-vaccination	[42]
6	Thailand and Philippines	1200 healthy children aged 12–18 months	JE-CV	Phase 3, randomized, observer-blind, active-controlled study	The overall seroprotection rate after a single JE-CV dose was 95.0%	[43]
7	Thailand	21 HIV-infected (13 male) and 101 HIV-uninfected (47 male) children aged ≥12 months, born to HIV-infected mothers	IMBV	Prospective study	Seroconversion rates were 83% (HIV-infected) and 99% (HIV-uninfected) one month after the second dose	[45]
8	India	360 healthy children aged 1–15 years	JE-VC (JENVAC)	Phase 4, multicenter, open-label, randomized controlled trial	81.7% seroprotection at 360 days post-vaccination	[46]
9	India	60 healthy children aged 1–3 years	JE-VC (IXIARO^®^)	Open-label randomized Phase II trial	Seroconversion rates were 95.7% (3 µg dose) and 95.2% (6 µg dose) at day 56 post-vaccination	[48]
10	Vietnam	250 healthy participants (aged 9 months–60 years)	JE-CV (IMOJEV^®^)	Prospective, open-label, single-center, single-arm study	A 4-fold increase in seroconversion from baseline was observed within 28 days post-vaccination (PV)	[50]
11	Philippines	571 infants aged 8 months (±2 weeks), ≥37 weeks gestation	LAJEV	Prospective, randomized, open-label, single-center study	JE seroprotection rates were 92.1% and 90.6% after LAJEV administration, including when co-administered with MV	[51]
**Effectiveness**						
1	Myanmar	198 school children aged 5–15 years	LAJEV	Cross-sectional descriptive study	All participants tested positive for anti-JEV IgG, and 87% developed NATs against JEV within 6 months	[37]
2	Thailand	152 healthy children (mean age 14.4 months)	JE-VC	Cross-sectional study	SPR was 89.3% one year after the primary series and 100% one month after the booster	[44]
3	Vietnam	JE patients (≤15 years old)	IMBV	Case–control study	Effectiveness was 92.9%	[47]
4	India	1075 healthy adults (≥15 years)	LAJEV	Observational study	Approximately 85.5% of participants developed antibodies within 28 days of vaccination	[49]

**Abbreviations:** LAJEV, live attenuated SA 14-14-2 vaccine; MV, measles vaccine; PV, post-vaccination; JE-CV, live attenuated chimeric JE vaccine; SPR, seroprotection rates; JE, Japanese encephalitis; PRNT50, plaque-reduction neutralization test; NATs, neutralizing antibodies titers; GMT, geometric mean titer; IMBV, inactivated mouse brain-derived JE vaccine; HIV, Human Immunodeficiency Virus; IgG, immunoglobulin G.

According to studies conducted in LMICs and summarized in Table 2, all types of JEVs—LAJEV, JE-CV, and inactivated vaccines have generally shown strong efficacy, though results vary by population and study design. Seroprotection rates (SPRs) for these vaccines were consistently reported to be above 80%. Although one study reported lower efficacy, the authors noted that this could be explained by various confounding factors Most studies in LMICs report seroprotection rates above 80% for different JE vaccines, though findings are not entirely consistent across settings.

Importantly, immunocompromised children, including those living with HIV, still achieved seroconversion rates (SCRs) above 80% after vaccination, despite the well-recognized challenges such individuals face in generating antibody responses. Booster doses of JEVs, regardless of the type of primary vaccine administered, have also been shown to be effective. This suggests that mixed primary–booster vaccination schedules using different vaccine types are feasible and immunogenic. Furthermore, JEVs have been effective in generating neutralizing antibodies (NATs) even when co-administered with measles vaccine (MV), which is particularly relevant during ongoing measles outbreaks in LMICs [52]. This finding suggests that simultaneous vaccination does not impair the immune response. In fact, co-administration of JEV with MV may improve adherence to vaccination programs by reducing the number of separate visits, while also ensuring earlier protection against both diseases [53]. Moreover, JEV can be administered alongside other routine pediatric vaccines without negatively affecting the immune responses of the co-administered vaccines [42].

In contrast, evidence from high-income countries (HICs) such as Australia, Japan, and Europe largely comes from non-endemic populations, including travelers and military personnel. These studies consistently show high seroprotection rates for JE-VC (IXIARO^®^) and JE-CV (IMOJEV^®^), with some cohorts maintaining >90% immunity up to 10 years post-vaccination [54,55]. This long-term immunogenicity data is less available in endemic LMIC populations, where studies typically assess protection within the first year after vaccination. The discrepancy highlights the need for extended follow-up in LMICs to better inform booster policies.

### 3.2. Safety

The safety profile of JEV is summarized below. Key findings from the included studies are presented in Table 3, which outlines observed adverse events (AEs), their frequency, and severity across study populations.

#### 3.2.1. LAJEV Safety Among Children and Infants

Studies on the safety of LAJEV in children and infants, ranging in age from 6 months to 18 years in endemic countries, consistently show that the vaccine does not cause significant adverse reactions [38,39,42,51,56,57]. Most reported AEs were mild, non–life-threatening, and resolved without deaths or long-term complications [39,42,51]. Although some serious adverse events (SAEs) were documented, these were not linked to vaccination and were classified as unsolicited reactions [39,40,48,53,56].

The mild AEs associated with LAJEV varied but most commonly included fever [42,48,51,57], irritability, hypersensitivity [56], swelling, local induration, pain, vomiting, insomnia, drowsiness, and diarrhea [51]. Among these, fever was the most frequently reported.

Some studies assessed the safety of LAJEV under specific conditions. For example, Capeding and colleagues conducted a randomized, open-label clinical trial evaluating the co-administration of LAJEV with the measles, mumps, and rubella (MMR) vaccine in infants. No redness at the injection site was observed in infants who received both vaccines, compared with those who received only MMR [42]. Similarly, Gatchalian and colleagues studied the co-administration of LAJEV and measles vaccine (MV) among infants in the Philippines and found only mild AEs [51]. These findings suggest that LAJEV remains safe even when given alongside other vaccines. The ability to administer vaccines simultaneously without unwanted interactions may improve adherence to vaccination schedules, reduce the number of visits, and facilitate the integration of new vaccines into national immunization programs [58].

#### 3.2.2. Inactivated JEV Safety Among Children

Studies of JE-VC reported fever as the most common AE, followed by mild symptoms such as vomiting and reduced appetite [44,46]. These events were self-limiting and did not require hospitalization. In a randomized controlled trial, Vadrevu and colleagues examined AEs following the interchange of CVI-JE and LAJEV. Although some SAEs were recorded, they were unrelated to vaccination and resolved within one week. No JE-related illness was reported during the three-year study period [46].

#### 3.2.3. JE-CV Safety Among Children and Infants

In studies of JE-CV in children and infants, fever was the most commonly reported AE, generally mild to moderate. Only one case of high fever (39.9 °C) was reported [41,43,50]. Injection-site reactions such as erythema, pain, swelling, and tenderness were also frequent [41,43]. Other AEs included loss of appetite, irritability, headache, malaise [43,50], uticaria [53], and a single case of diarrhea [50]. Systemic AEs, dermatitis, and syncope were deemed unrelated to vaccination [41,50]. Available data suggest that JE-CV has a favorable safety profile, with no vaccine-related deaths or withdrawals reported in the reviewed studies [41,50,53]. Safety data on IMBV in pediatric populations also indicated mostly mild reactions, such as injection-site pain, headache, nausea, and vomiting. These findings were consistent even among children with immunosuppressive conditions, including HIV [45].

#### 3.2.4. JEV Safety Among Adults

Safety data for LAJEV in adult populations in LMICs are limited. The most common AEs were injection-site pain, erythema, and swelling. During a 28-day post-vaccination follow-up, mild symptoms such as rash, irritability, nasopharyngitis, and fever were reported. Two SAEs—dermatitis and syncope—were classified as unrelated to the vaccine [49]. For JE-VC, systemic reactions such as fever, headache, and appetite loss varied by age group. One participant developed grade 3 erythema (>10 cm in diameter), and another experienced grade 2 fever. However, no vaccine-related SAEs or withdrawals occurred, and only one mild AE (diarrhea) was considered possibly vaccine-related. Overall, moderate or severe reactions were rare, and the vaccine demonstrated an acceptable safety profile [50].

Safety findings from HICs are broadly consistent with LMIC data. Post-marketing surveillance in Australia and the European Union has reported low rates of systemic or local adverse events for JE-VC and JE-CV, comparable to those observed in LMIC trials [59,60]. However, AE monitoring systems in HICs are generally more robust, underscoring the importance of strengthening pharmacovigilance capacity in endemic LMIC settings.

**Table 3 vaccines-13-01038-t003:** Safety Profile of JEV.

No	Country Site	Participant	Type of JE Vaccine	Type of Study	Result	Ref.
1	Bangladesh	Children aged 3–4 years	LAJEV	Phase IV, open-label clinical trial	No vaccine-associated neurological AEs or SAEs were observed after the booster dose	[38]
2	Sri Lanka	Healthy children aged 2–5 years	LAJEV	Open-label, single-arm trial	Unsolicited adverse reactions were reported in 9.9% of 2-year-olds and 9.7% of 5-year-olds	[39]
3	Sri Lanka	Healthy infants aged 9 months (±2 weeks)	LAJEV	Open-label, non-randomized, single-arm trial	Most adverse reactions were mild, and no SAEs were related to vaccination	[40]
4	Thailand	50 healthy children aged 1–5 years (64% male)	JE-CV	Open-label clinical trial	Mild injection-site reactions: erythema (18%), pain (10%), swelling (4%). Systemic reactions included fever (8%) and URTI (16%)	[41]
5	Philippines	628 healthy Filipino children aged 9–12 months	LAJEV	Randomized clinical trial	No cases of post-vaccinal encephalitis or vaccine-related SAEs were reported in either group	[42]
6	Thailand and Philippines	1200 healthy children aged 12–18 months	JE-CV	Phase 3, randomized, observer-blind, active-controlled study	Solicited reactions were reported in 66.5% of participants, while unsolicited AEs were infrequent (1.2%) and mild	[43]
7	Thailand	152 healthy children (mean age 14.4 months)	JE-VC	Cross-sectional study	Self-limiting symptoms were observed within 28 days post-vaccination: fever (17.6%), vomiting (8%), and poor appetite (5.3%)	[44]
8	Thailand	21 HIV-infected (13 male) and 101 HIV-uninfected (47 male) children ≥ 12 months old	IMBV	Prospective study	AEs occurred in 32% of HIV-infected and 31% of HIV-uninfected children (*p* = 0.82)	[45]
9	India	360 children randomized equally	JE-VC (JENVAC)	Phase IV, multicenter, open-label, randomized controlled trial	At least one AE was reported by 57 children in the JEN-VAC group and 62 in the LAJEV group. Fever was the most common solicited AE: 39.1% (JEN-VAC) vs. 28.6% (LAJEV)	[46]
10	India	60 healthy children aged 1–3 years	JE-VC (IXIARO^®^)	Open-label randomized phase II study	13 AEs were reported in 12 subjects (*p* = 0.29)	[48]
11	India	1075 healthy adults (≥15 years old)	LAJEV	Observational study	Four participants reported minor symptoms within 28 days	[49]
12	Vietnam	250 healthy participants aged 9 months–60 years	Live-attenuated chimeric vaccine (IMOJEV^®^)	Prospective, open-label, single-center, single-arm study	A single dose elicited a protective immune response and was well tolerated, with no safety concerns	[50]
13	Philippines	Infants < 1 year of age	LAJEV	Prospective, randomized, open-label, single-center study	The most common systemic reactions at day 0 were irritability and drowsiness. Irritability, drowsiness, and insomnia persisted at low rates until day 3.	[51]
14	Thailand	10,000 healthy children aged 9 months–<5 years	JE-CV	Phase IV, prospective, open-label, multicenter study	SAEs occurred in 204 participants (3.0%) in Group 1 and 59 participants (1.9%) in Group 2. Of 294 SAEs in 263 participants, only 3 events in 2 participants were considered vaccine-related.	[53]
15	Cambodia	~310,000 children aged 9 months–12 years	LAJEV	Observational safety surveillance study	28 AEFIs occurred, giving an incidence of 9.0 per 100,000 doses. Most common events were vasovagal episodes (7 cases, 25%) and rash (6 cases, 21%); others were typical childhood illnesses such as fever and vomiting.	[56]
16	Sri Lanka	3041 infants vaccinated at 9 months	LAJEV	Cohort	Of 2878 infants followed for 14 days, 911 (32%) experienced 1423 AEFIs. Among these, 376 (26%) were causally linked to LAJEV. Most common were irritability (53/1000 doses) and fever ≥ 100.4 °F (46/1000 doses)	[57]

**Abbreviations**: LAJEV, live attenuated SA 14-14-2 vaccine; SAE, serious adverse events; JE-VC, inactivated vero cell culture vaccine; PV, post-vaccination; IMBV, inactivated mouse brain-derived JE vaccine; HIV, Human Immunodeficiency Virus; AE, adverse events; JE-CV, live attenuated chimeric JE vaccine AEFI, adverse event following immunization; URTI, Upper Respiratory Tract Infection.

### 3.3. Cost of Japanese Encephalitis Vaccination Program

Multiple studies in LMICs, including Indonesia, Bangladesh, Cambodia, India, and the Philippines, have demonstrated the cost-effectiveness of JE vaccination programs through Markov models and retrospective economic evaluations from both governmental and societal perspectives (Table 4). Across settings, vaccination was associated with significant health gains and economic savings, with incremental cost-effectiveness ratios (ICERs) consistently below national GDP Per Capita thresholds.

In Indonesia, Kosen and colleagues used a Markov model to assess several strategies, concluding that a national routine immunization program could avert 11,149 JE cases, 2564 deaths, and 78,349 DALYs across three birth cohorts. This strategy was also cost-saving, with estimated savings of $9.1 million from the government perspective and $33.7 million from the societal perspective [61]. Additional modeling by Liu and colleagues confirmed cost-effectiveness, estimating $31 per DALY averted [62]. Putri and colleagues retrospectively compared routine versus campaign-based immunization, finding routine vaccination to be more cost-effective [63].

In Bangladesh, Nguyen and colleagues found that both a subnational campaign with subnational routine immunization (S1) and a subnational campaign with national routine immunization (S2) were cost-effective, averting up to 13,176 cases and 2635 deaths. In contrast, a national routine-only strategy (S3) prevented fewer cases (9876) and deaths (1973) and was less efficient [64]. In the Philippines, Vodicka and colleagues conducted a Markov model analysis showing that the most cost-effective approach was a national campaign followed by routine delivery, with costs per DALY averted of $233 (government perspective) and $29 (societal perspective). This program could prevent 27,856–37,277 JE cases and 5571–7455 deaths at an incremental cost of $45.9–53.9 million [65].

In India, Singh conducted a Cost–Benefit analysis of LAJEV vaccination, reporting a discounted net benefit of ₹598.52 million ($9.82 million) and an internal rate of return (IRR) of 291% [66]. In Cambodia, Touch and colleagues assessed costs and cost-effectiveness, finding the most favorable strategy to be a campaign targeting children aged 1–10 years, followed by routine vaccination at 9 months. Over 15 years, this approach was projected to prevent 2888 cases, 376 deaths, and 2354 disabilities, reducing the burden by 52,392 DALYs and saving up to $1.46 million. The ICER ranged from $21.84 to $63.71 per DALY averted, well below Cambodia’s 2008 GNI Per Capita of $723. Sensitivity analyses to test the robustness of the cost-effectiveness results under varying assumptions confirmed that the intervention remained cost-effective, particularly when the cost per DALY averted exceeded $7 [67]. Taken together, these studies suggest that JE vaccination is generally cost-effective across diverse LMIC settings, although results depend on assumptions, vaccine type, and local context.

Economic evaluations in HICs differ substantially from LMIC findings. In non-endemic countries, JE vaccination is typically assessed for travelers, military personnel, or laboratory staff, and population-level programs are generally not considered cost-effective due to the very low incidence of disease [68]. By contrast, in endemic LMICs, vaccination is consistently cost-effective at the national level, with incremental cost-effectiveness ratios well below GDP Per Capita thresholds [68,69]. This divergence illustrates how epidemiological context strongly shapes economic value.

**Table 4 vaccines-13-01038-t004:** Cost of Japanese Encephalitis Vaccination Program.

No	Country	Study Type	Vaccine Type	Cost Findings	Conclusion	Ref.
1	Indonesia	CEA, Markov model	N/A	At the national level, a one-time campaign followed by routine immunization across three birth cohorts could prevent 31,386 JE cases, 7219 deaths, and 231,234 DALYs, at an additional cost of $68.7 million (government) and $39.8 million (society). At the subnational level, the same strategy could prevent 4099 cases and 943 deaths, with an additional cost of $10.2 million (government) and $5.9 million (society). ICERs per DALY averted remained cost-effective: $297 (government) and $172 (society) nationally, and $322 (government) and $197 (society) subnationally	Compared with no vaccination, JE vaccination would avert more DALYs at lower marginal costs and could even result in savings by offsetting treatment costs	[61]
2	Indonesia	CEA	LAJEV	Routine JE vaccination could prevent 54 cases and 5 deaths, saving 1224 healthy life-years. The cost was estimated at $700 per case prevented and $31 per DALY averted	Highly cost-effective; supports routine JE immunization in endemic areas	[62]
3	Bangladesh	CEA, Markov model	LAJEV	Subnational campaign plus routine immunization (S1) cost $82.2 million over 20 years, saved $75.1 million in healthcare costs, and prevented 7554 cases and 1509 deaths. S1 had low ICERs and was cost-effective in 99% of simulations. Subnational campaign with national routine immunization (S2) prevented 13,176 cases and 2635 deaths, saving $134.8 million but costing $154 million. The national routine-only strategy (S3) prevented 9876 cases and 1973 deaths, saving $104.8 million but was least efficient.	JE vaccination is cost-effective in Bangladesh. S1 was the most cost-effective; S2 provided greater health benefits at higher cost; S3 was the least efficient	[64]
5	Indonesia	Retrospective economic analysis, CEA	LAJEV	Routine JE vaccination was the most cost-effective strategy, with a cost per DALY averted of $212.59 (government) and $94.09 (society). By contrast, the vaccination campaign and introduction approach cost $1473.53 (government) and $1224.20 (society) per DALY averted)	Both vaccination strategies were cost-effective, but neither was cost-saving compared with no immunization	[63]
6	India	Cost–benefit study	LAJEV	JE vaccination generated a total discounted net benefit of ₹598.52 million ($9.82 million), with an internal rate of return (IRR) of 291%. Each rupee invested could yield ₹11 in benefits per person over five years	Despite operational challenges, JE vaccination is a sound investment	[66]
7	Cambodia	CEA	LAJEV	The average societal cost per JE case was $441.05, including $41.16 in pre-hospital costs, $308.83 in hospitalization costs, and $78.89 in post-discharge expenses. A combined strategy (campaign for children aged 1–10 years followed by routine immunization at 9 months) could avert 2888 cases, 376 deaths, and 2534 disabilities, saving $1.46 million and reducing the burden by 52,392 DALYs. ICERs ranged from $21.84–$63.71 per DALY averted, well below Cambodia’s 2008 GNI Per Capita of $723	LAJEV is highly cost-effective in Cambodia, even under uncertainty	[67]
8	Philippines	CEA, Markov Model	LAJEV	A national campaign followed by routine immunization was the most cost-effective strategy, with costs per DALY averted of $233 (government) and $29 (society). National-only or subnational campaign-plus-routine strategies showed similar cost-effectiveness ($233–$265 government; $29–$57 society). Overall, vaccination could prevent 27,856–37,277 cases, 5571–7455 deaths, and 173,233–230,704 DALYs in children <5 over 20 cohorts. Additional costs were $45.9–$53.9 million ($230,000–$440,000 per year government; $6.6–$9.8 million society).	LAJEV in the Philippines is expected to be cost-effective, reducing long-term costs and improving health outcomes compared with no vaccination	[65]

**Abbreviations:** CEA, cost-effectiveness analysis; JE, Japanese encephalitis; DALY, disability-adjusted life year; ICER, incremental cost-effectiveness ratio; LAJEV, live attenuated SA 14-14-2 vaccine; GDP, gross domestic product; IRR, internal rate of return; GNI, gross national income; N/A, not applicable. **Note:** Several studies (e.g., from Bangladesh, the Philippines, and certain Indonesian analyses) are model-based projections, whereas others (e.g., from Cambodia, India, and retrospective Indonesian evaluations) are derived from implemented vaccination programs.

### 3.4. Policy on Japanese Encephalitis Vaccination Program

In Bangladesh, three JE immunization strategies have been evaluated. Strategy 1 (S1) involves a subnational one-time campaign for children under 15 years, combined with subnational routine immunization at 9 months, targeting high-incidence provinces (Rangpur, Rajshahi, Chattogram). Strategy 2 (S2) consists of a subnational campaign plus national routine immunization across all provinces. Strategy 3 (S3) is limited to national routine immunization only. Among these, S1 had the lowest overall cost ($82.2 million) and the greatest health impact, preventing 7544 cases and 1509 deaths. It was also the most cost-effective, with ICERs of $94/DALY averted (societal perspective) and $981/DALY averted (government perspective). Thus, S1 was identified as the optimal strategy for Bangladesh [64].

In Indonesia, immunization policy is structured at both national and subnational levels. The national program covers all 34 provinces and offers two options: (1) a one-time campaign for children aged 1–15 years, followed by routine immunization for one birth cohort and three subsequent cohorts, or (2) routine infant immunization only. Subnational programs in seven high-risk provinces also follow two similar options. Since children under 15 years account for more than 85% of JE cases in Indonesia, these strategies were designed to maximize public health impact. A national routine infant immunization strategy alone was estimated to prevent 11,149 JE cases, 2564 deaths, and 78,349 DALYs, while saving $9.1 million (government perspective) and $33.7 million (societal perspective) [61].

In China, Liu and colleagues reviewed JE vaccination policies through stakeholder interviews and document analysis across eight provinces. Four provinces (Group A) with EPI support recommended four doses of inactivated vaccine or three doses of LAJEV for preschool-aged children. Group B provinces, which lacked EPI participation, showed greater variation, with some requiring vaccination up to age 6 or reducing LAJEV schedules from three to two doses due to adequate efficacy. Financing relied on government funding, EPI insurance, and family out-of-pocket payments. EPI vaccines were provided free of charge, with provincial governments funding service delivery. Implementation of JE immunization dramatically reduced incidence: in Group A provinces, cases dropped from 18.6 per 100,000 in the 1960s to 0.7 per 100,000 in the 1990s (96.2% reduction). In Group B provinces, incidence declined from 9.6 to 15.6 per 100,000 in the 1960s–70s to 3.3 per 100,000 in the 1990s [70].

Policy approaches in HICs also contrast with those in LMICs. For example, Australia recommends JE vaccination primarily for travelers and residents in affected northern regions, while Japan historically integrated JE vaccination into its national immunization program with high coverage [71,72]. These strategies differ from LMIC contexts, where programs must balance high endemic burden with limited financing and infrastructure. The comparison underscores how health-system capacity and epidemiology jointly shape policy adoption [72,73].

## 4. Discussion

### 4.1. Efficacy, Effectiveness, Safety

Between 2005 and 2025, studies in LMICs reported the use of four JEV types: LAJEV, JE-CV, JE-VC, and IMBV. Most research has focused on infants and children, while evidence on adults remains limited. The majority of studies demonstrated high efficacy and effectiveness, with seroconversion rates (SCRs) frequently exceeding 80% and surpassing the ≥75% threshold. However, two LAJEV studies conducted in the Philippines and Sri Lanka reported SCRs below 75%, which contrasts with WHO’s 2015 Position Paper that recommended LAJEV among the available vaccines [24]. These discrepancies may partly reflect differences in the laboratories performing neutralizing antibody assays [42].

Although IMBV showed >90% effectiveness and SCRs above the acceptable threshold in studies from Vietnam and Thailand, WHO recommends replacing IMBV with vaccines such as LAJEV, JE-VC, or JE-CV [24]. While IMBV remains effective in some countries, it is less favorable due to safety concerns, higher costs, variability in manufacturing, and the requirement for multiple doses and boosters. The discrepancy likely reflects timing: WHO’s position paper was published in 2015, whereas the IMBV studies in Vietnam [47] and Thailand [45] were conducted before or around that period. More recent studies in these countries align with WHO’s updated recommendations [41,43,50]. This suggests that LMICs have progressively adopted vaccines consistent with WHO’s most recent guidance.

In non-endemic HICs such as Australia, IMBV historically showed poor immunogenicity: in 2005, only 32% of vaccinated individuals demonstrated protective antibodies, with 37% of adults immune compared to just 24% of children. Local and systemic adverse events were common (10–30%), including fever, headache, myalgia, malaise, and injection-site reactions in approximately 20% of recipients [74]. Today, Australia has replaced IMBV with two alternatives: IMOJEV (JE-CV) and JEspect (JE-VC). In a cross-sectional study, Mills and colleagues found that all adults vaccinated within the past five years remained seropositive, and 92.9% of those vaccinated more than five years earlier remained protected, suggesting durable immunity exceeding 90% up to 10 years post-vaccination. Safety outcomes were not reported in that study. However, a post-marketing safety study in Australia comparing adverse events following immunization (AEFIs) of IMOJEV and JEspect (IXIARO) found low overall rates: 7.0% for any event, 2.8% for systemic events, and 1.9% for local events, with no significant differences between the two vaccines [75].

These findings are broadly consistent with LMIC evidence, where JE-CV and JE-VC have been reported as safe and well tolerated in most studies. However, long-term immunogenicity data in LMICs remain scarce, as most studies only track responses within 7–360 days after vaccination. Because many LMICs are endemic regions (e.g., India, Sri Lanka, Vietnam, Myanmar, Thailand, the Philippines), natural exposure to the virus may provide periodic immune boosting. Nevertheless, there is a clear need for long-term immunogenicity studies to guide vaccine policy and optimize dosing schedules for durable protection. WHO’s 2015 Position Paper also highlights this research gap [24]. Additionally, there is limited evidence on vaccine safety and efficacy in special populations in LMICs, including pregnant women and immunocompromised individuals, underscoring the urgent need for further research.

Taken together, the evidence suggests that JE vaccines consistently achieve protection above global benchmarks, with the notable exception of IMBV, which is now being phased out. The consistency of findings across multiple vaccine types and settings supports the role of JE vaccination as an important strategy for encephalitis control in LMICs, though further evidence is still needed in some areas (e.g., adults, long-term follow-up).

### 4.2. Cost

Evidence consistently shows that JE vaccination is highly cost-effective across diverse LMIC settings, with incremental cost-effectiveness ratios (ICERs) well below GDP Per Capita thresholds. Although methods differ, both model-based projections and real-world evaluations confirm substantial health and economic benefits.

From a governmental perspective, JE vaccination programs are highly cost-effective, with the Philippines demonstrating the strongest results. Vodicka and colleagues reported that a national campaign followed by routine immunization using LAJEV averted cases at a cost of just $233 per DALY from the government perspective and $29 per DALY from the societal perspective. This strategy prevented up to 37,277 cases and 7455 deaths with an investment of $45.9–53.9 million [61]. The Philippines’ success is attributed to the use of low-cost LAJEV, broad program implementation, and a high disease burden, which maximized health impact and economic return. While Indonesia and Bangladesh also achieved positive outcomes, the Philippines achieved the most favorable balance of health gains and cost efficiency.

A multicriteria decision analysis (MCDA) in Bangladesh further supports JE vaccination as a priority. In this study, JE was compared with six other vaccine-preventable diseases using both quantitative and qualitative criteria. Despite JE’s relatively low incidence, its high fatality rate (~30%), strong vaccine efficacy (>90%), and cost-effectiveness led stakeholders to rank JEV as the top candidate for national vaccine introduction [76].

Several methodological points should be considered. The evidence includes both evaluations of implemented programs (e.g., Cambodia, India, selected regions of Indonesia) and hypothetical cost-effectiveness projections (e.g., Bangladesh, the Philippines, and modeling studies in Indonesia). Thus, results should be interpreted with caution, as not all reflect active national immunization policies. From a societal perspective, non-medical costs such as productivity loss, transportation, and post-discharge care substantially influence cost-effectiveness. From a governmental perspective, routine immunization is generally more cost-saving than campaign-only strategies. Disease burden, vaccine effectiveness, unit cost, and geographic coverage also strongly affect economic outcomes.

However, these findings are constrained by several limitations. Many cost-effectiveness studies rely on modeling assumptions rather than real-world program data, which may not fully capture delivery challenges. Furthermore, most analyses focus on children, with little evidence on adults, older populations, or special groups such as pregnant women. Differences in incidence estimates, vaccine pricing, and costing methods across studies also reduce comparability between countries.

Overall, JE vaccination consistently emerges as a cost-effective public health intervention, frequently leading to net cost savings, particularly when LAJEV is used. Combined strategies (routine plus campaign) deliver the greatest health benefits, though sometimes at higher cost. Despite methodological variation, the overall evidence base strongly supports JE vaccination in endemic LMICs from both government and societal perspectives.

### 4.3. Policy

JE vaccination policies in LMICs, such as Bangladesh and Indonesia, share similarities with approaches in developed countries like China. Across these settings, policies emphasize tailored strategies to maximize health outcomes in specific epidemiological and resource contexts.

In Bangladesh, the most cost-effective option is Strategy 1 (S1), a subnational program that combines routine vaccination with a one-time campaign in high-incidence provinces. This approach prevented 7544 cases and 1509 deaths at a cost of $82.2 million. In Indonesia, policies integrate both national and subnational strategies, with routine infant immunization as a central feature. Over time, this program has prevented 11,149 cases and 2564 deaths while generating substantial cost savings. In China, policies differ between affluent and less-developed provinces. The national program recommends four doses of inactivated vaccine or three doses of LAJEV for preschool-aged children, but provinces adapt this guidance: some require up to six doses, while others reduced LAJEV from three to two doses due to adequate efficacy. Financing is equally diverse, involving a mix of national government funding, EPI insurance, and out-of-pocket contributions.

These findings underscore that JE vaccination policies are most effective when adapted to local epidemiology, health system capacity, and financing structures. Strategic targeting of high-risk populations, integration into routine immunization, and flexible financing are essential to maximizing benefits while minimizing costs.

Future research should prioritize long-term immunogenicity studies in endemic LMICs to inform optimal dosing and booster schedules. Cost-effectiveness analyses incorporating indirect benefits—such as productivity gains, equity impacts, and climate-driven changes in JE transmission—would also strengthen the evidence base. Finally, head-to-head evaluations of LAJEV, JE-CV, and JE-VC in real-world programs are urgently needed to guide national policy decisions.

Taken together, these insights suggest that JE vaccination policies, when context-specific and sustainably financed, can deliver both substantial health gains and economic value, reinforcing JE vaccination as a cornerstone of public health in endemic regions.

## 5. Conclusions

Currently available JE vaccines in LMICs demonstrate high efficacy and effectiveness across vaccine types and age groups, with the exception of IMBV, which WHO no longer recommends due to safety concerns. Safety profiles are favorable, with mostly mild adverse events, fever being the most common. JE vaccination is also highly cost-effective—particularly with LAJEV—delivering substantial health and economic benefits, especially when supported by policies tailored to national circumstances and backed by sustainable financing. Comparisons with high-income countries (HICs) highlight both similarities and differences. Vaccination in HICs is generally targeted to travelers, military personnel, or small regional populations, producing robust long-term immunogenicity and safety data. By contrast, LMIC programs serve endemic populations where vaccination is essential for control, highly cost-effective, but challenged by financing, infrastructure, and limited long-term evidence. In summary, JE vaccination remains a safe, effective, and cost-effective intervention in LMICs. Sustained investment, context-specific policies, and stronger long-term studies are critical to optimize impact and reduce inequities in vaccine-preventable diseases between LMICs and HICs.

## Figures and Tables

**Figure 1 vaccines-13-01038-f001:**
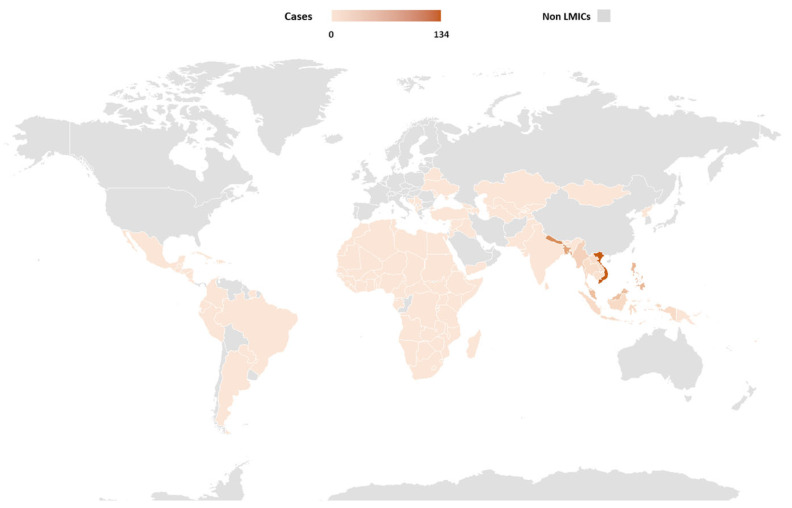
Geographical Distribution of JE in 2024 (Adapted from WHO [20]). **Abbreviations:** Non-LMICs, Non-Low-Middle-Income Countries.

**Table 1 vaccines-13-01038-t001:** Summary of JEV, Schedules, and Target Groups.

Vaccine Type	Schedule	Target Group	Ref.
IMBV	Two doses administered 28 days apart (primary series). A booster dose is given one year after completion of the primary series.	Children age 1–2 years	[23]
LAJEV	Single dose	Infants ≥8 months of age	[24]
JE-VC	Two doses administered 28 days apart (primary series, schedule may vary by manufacturer). The first booster dose is given one year after the primary series, and the second booster at 5–6 years of age	Infants ≥6 months of age in endemic settings	[24]
	Two doses administered 28 days apart (primary series). A booster dose may be considered one year after the primary series.	Adults	[25]
JE-CV	Single dose	Individuals aged ≥9 months to <18 years	[26]

**Abbreviations:** LAJEV, live attenuated SA 14-14-2 vaccine; JE-CV, live attenuated chimeric JE vaccine; IMBV, inactivated mouse brain-derived JE vaccine; JE-VC, Inactivated Vero cell culture vaccine.

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
