# Peer review of "Japanese Encephalitis Vaccine in Low- and Middle-Income Countries (LMICs): A Narrative Review of Efficacy, Effectiveness, Safety, Cost, and Policy"

_vaccines, 2025, doi:10.3390/vaccines13101038_

Round 1

Reviewer 1 Report

Comments and Suggestions for Authors

The review from Pilihanto et al aims to provide an overview of Japanese Encephalitis (JE) vaccines in low- and middle-income countries (LMICs), covering efficacy, effectiveness, safety, cost, and policy considerations though a review of the literature from 2005 to 2025.   The aim is to summarise what is currently know. but the impact of review is somehow limited by the lack of comparison with the high-income countries (HICs) regarding JE vaccine efficacy, effectiveness, safety, cost, and policy. Including comparative insights from HICs—particularly in the conclusion—could enhance the review’s relevance by highlighting contextual similarities or differences. Such a comparison it is indeed done in the policy section using China as comparator, but it should be more substantial elsewhere.

The review requires a substantial revision as the main issue is that the references appear to be out of sync and may not be correctly cited, there is difficult as a reviewer to check whether the sentences are supported by the data.  These are some examples from the Policy section but issues starts earlier:

Line 552: paragraph is about policy strategy in Bangladesh and the ref 52 is about cost-effectiveness in Bali, Indonesia.

Line 569: paragraph about JE cost in Indonesia and ref 49 is about the result of phase IV clinical trail in Thailand.

Later on, on the policy strategy in China (570-593), no reference seems to be there to support the statements. There is  ref 57 but that is about policy settings in Bangladesh.

Also, references 14 and 15 are the same, and there is a formatting issue for ref 21-22.

The use of tables is effective in summarising the key findings of the studies (although references should be checked) but the main body of the text section 3, 4 and 5, appears a bit repetitive, particularly as methods and results are already summarised in the tables. To strengthen the manuscript, the text should highlight trends, interpret finding, add extra details and compare studies. Discuss the broader implications of the findings and synthesise the results to highlight key similarities and differences across vaccine types and studies. This would help create a more cohesive and insightful narrative tailored to LMIC settings.

Another major issue is on Efficacy and effectiveness of vaccines paragr. 3. The authors mentioned that neutralising antibody titres are used to determine vaccine efficacy. They should explain the reason behind this, as a vaccine efficacy is usually determined based on protection from infection in large clinical trials.  The use of Neutralising antibody titres as correlate/surrogate of protection needs to be introduce with the correct reference. The authors mention WHO guidelines and ref 25 is on SARS-CoV-2 not JEV. The correct reference should be: Hombach, J., Solomon, T., Kurane, I., Jacobson, J. & Wood, D. Report on a WHO consultation on immunological endpoints for evaluation of new Japanese encephalitis vaccines, WHO, Geneva, 2–3 September, 2004. Vaccine 23, 5205–5211 (2005).     

I am not convinced that the review achieved what was aimed to. There is no information on how many vaccines are currently in use, in which countries they are licensed.  A summary of the main vaccines, manufacturer, type of vaccine (LA, inact.chimeric etc) in which country they are licensed will be helpful to follow the review.
Also, figure 1 could be improved and it may be more useful to present the total number of JE cases reported in 2024 and then describe the geographical distribution and its changes over time within the main body of the text. This approach would allow for a clearer and more focused visual representation, complemented by a narrative discussion of spatial and temporal trends. Figure 1b. It is not clear from the text the reason to have the 2001-2024 map versus the 2024 as there are very few changes. More interestingly would be the distribution of the different genotypes in those countries.

Minor points:

Line 37: Flaviridae FAMILY.

Line 38: wild bird – specifically aquatic wading birds

Line 42: “but recent updates indicate that adults can also be susceptible. In endemic regions, children are mostly affected due to adult immunity.”  The sentences need rewording. What the authors mean is that majority of cases are seen in children younger than 15 because adults in an endemic country have already been exposed to and acquired immunity.

Line 51: symptomatic:asymptomatic cases 1:25-1000 is a big range, where this data come from? WHO reckons that is 1:250

Line 52: 30% mortality in those who have symptoms

Line 80: JE is 1.8 per 100,000 resulting in 10,000-15000 deaths. Authors should specify what 100,000 are (i.e. individuals). These data are from references 16 and 17 which are dated 2009 and 2011 respectively. A more up-to-date estimate can be found in WHO website with an estimate  100 000 clinical cases (95% CI: 61 720–157 522) of JE globally each year, with approximately 25 000 deaths (95% CI: 14 550–46 031). Update data with more recent references.

Same for the next paragraph.

Line 88 add a reference to the Australian news as I struggle to find it

Line 96. By whom? Add reference. Is it the same as line 106-107?

Line 97-98 are repeated in line 98-99.

It will be interesting to know how many articles were found from the literature search.

Definition of efficacy and effectiveness should be provided.Also, acronyms are not always spelled at their first use.

Line 128: needs rewording: the limited efficacy assessment in adults in LMIC is because there are endemic countries and therefore acquired immunity as children.

Table 2 is not mentioned in the text. It should be introduced at the beginning of section 4. The text is very long and the table does a good job in summarizing it. Rather than repeat what is in the table, in the text it will be more useful to summarize the side effects rather than study by study. The most important question is are the safety data different from the one observed conducting study in the non-endemic regions? Are there any extra risks?

Line 449: China is listed as a LMIC although later, correctly is mentioned as a developed country.

Line 501 what are sensitivity analyses?

Table 3 contains a summary of the literature on the cost effectiveness of JE strategies. It is not clear whether these countries have an active immunization policy for JE or are all hypothetical?

Line 526-528. Reword sentence as it is not cleat what the meaning is

577: couldn’t check the reference, is this correct? 4 doses of inactivated JE or 3 of the live attenuated? I thought most JE vaccines only require 2 doses?

Comments on the Quality of English Language

as mentioned above, some sentences need rewording as the message is not clearly explained

Reviewer 2 Report

Comments and Suggestions for Authors

The review manuscript “Japanese Encephalitis Vaccine in Low- and Middle-Income Countries (LMICs): A Narrative Review of Efficacy, Effectiveness, Safety, Cost, and Policy” provides a comprehensive overview of studies conducted in LMICs about different aspects of JEVs. This is an important endeavor as recent epidemiological studies indicate a spread of the virus from endemic areas to other regions that may become endemic in the future. Since the reservoir hosts of the JEV are birds and amplifying hosts are swine, the virus has definitely the potential to spread wherever the culex vector can survive. The manuscript is well organized but there are a number of shortcomings as detailed below.

General remarks:

Acronyms and abbreviations should be written out at first appearance in the main text. There are numerous occasions where this rule was not obeyed.

In all tables acronyms and abbreviations must be explained in a footnote. AS a service to the reader provide next to the type of the vaccine the brand name in parentheses.

While the English is in general of good quality, there are many cases where either a definite or indefinite article is missing. Thorough proof-reading is recommended.

The terms effectiveness and efficacy should be used consistently. Effectiveness refers to the observation of protection in a certain population under real-world conditions, while efficacy refers to the effect observed in a controlled trial.

Specific remarks:

  1. The vaccine IXIARO™ is not a live-attenuated vaccine. It is an inactivated vaccine propagated in vero cells.

  1. Introduction: Insert at a suitable location a paragraph relating that there are five genotypes of the virus and specify which vaccines are derived from which genotype. Furthermore, since genotype V seems to be reemerging include a statement about cross-protection.

  1. Line 38: After ‘mosquitoes’ there is instead of a comma an apostrophe.

  1. Lines 41ff: JE is not a childhood disease. It only appears as a childhood disease in endemic regions because adults are immune. Please reformulate the passage as you pointed yourself to this misinterpretation.

  1. Lines 97ff: This passage needs to be rewritten because the different JE vaccines have different schedules and different target groups. Specify for all four vaccines the schedule and the groups they are licensed for.

  1. Line 173: Delete the second ‘vaccine’

  1. Lines 181/182: The reason for the lower seroprotection rates in this study are most likely differences in the neutralization test and not unmeasured factors.

  1. Line 195: There is not an 96.4% increase. The increase is more than 14-fold!

  1. Line 205: Delete “geometric”

  1. Line 231: The age-related decrease of seroconversion rates is an artifact of the definition of seroconversion. It is biologically impossible to mount a fourfold titer increase if the titers are already very high as occurs in endemic areas with increasing age.

  1. Lines 238ff: The text here is misplaced. As mentioned above, all the details about the available vaccines should be placed into the Introduction.

Comments on the Quality of English Language

As mentioned in my report to the authors, quality of English is generally good, however, there are often omissions of required definite and indefinite articles. I have suggested thorough proof-reading.

Reviewer 3 Report

Comments and Suggestions for Authors

The manuscript addresses an important public health issue, namely the use of Japanese Encephalitis (JE) vaccines in low- and middle-income countries (LMICs). Despite it addressing an important issue, it requires significant improvements in methodology, synthesis, and critical discussion to meet the standards of a peer-reviewed narrative review.

Major comments

  1. Although this is a narrative review, the methodology section would benefit from greater clarity regarding how studies were identified and selected (e.g., databases searched, time frame, key inclusion criteria). This would help readers better understand the scope and potential limitations of the evidence summarized.
  2. The review presents detailed results from individual studies, often reproducing numerical data extensively. This leads to a text-heavy narrative. A more synthetic approach (e.g., summary tables, comparative analyses, highlighting patterns and gaps) would make the manuscript more readable and impactful.
  3. Distinction between efficacy and effectiveness is not consistently maintained. The terms are sometimes used interchangeably, which may confuse readers. Please revise carefully.
  4. The manuscript lacks a dedicated Discussion section. While interpretative remarks are scattered throughout the results sections, there is no synthesis of key findings, critical appraisal of the evidence, or comparison with previous reviews and WHO guidance. A structured Discussion would strengthen the manuscript considerably by contextualizing the findings, highlighting gaps in knowledge, and outlining implications for research and policy

Minor Comments

  1. Language and grammar require revision throughout for clarity and conciseness. For example, “effectivity” should be replaced with “effectiveness.”
  2. There are redundancies (e.g., recommended JE vaccine schedule is repeated in different sections). Streamlining would improve flow.
  3. The reference list should be updated to include more recent systematic reviews, WHO position papers, and global burden estimates (e.g., post-2022 data).
  4. Abbreviations should be defined once and used consistently (e.g., LAJEV, JE-CV, IMBV).
  5. Formatting inconsistencies are present in tables and text (e.g., spacing, capitalization of vaccine names).

Round 2

Reviewer 3 Report

Comments and Suggestions for Authors

Authors have appropriately addressed the comments.